# Precision Balance Assessment in Parkinson’s Disease: Utilizing Vision-Based 3D Pose Tracking for Pull Test Analysis

**DOI:** 10.3390/s24113673

**Published:** 2024-06-06

**Authors:** Nina Ellrich, Kasimir Niermeyer, Daniela Peto, Julian Decker, Urban M. Fietzek, Sabrina Katzdobler, Günter U. Höglinger, Klaus Jahn, Andreas Zwergal, Max Wuehr

**Affiliations:** 1German Center for Vertigo and Balance Disorders (DSGZ), LMU University Hospital, LMU Munich, 81377 Munich, Germany; ntnellrich@googlemail.com (N.E.); niermeyer.kasimir@googlemail.com (K.N.); d.peto@campus.lmu.de (D.P.); deckerjulian@icloud.com (J.D.); klaus.jahn@med.uni-muenchen.de (K.J.); max.wuehr@med.uni-muenchen.de (M.W.); 2Schoen Clinic Bad Aibling, 83043 Bad Aibling, Germany; 3Department of Neurology, LMU University Hospital, LMU Munich, 81377 Munich, Germany; urban.fietzek@schoen-klinik.de (U.M.F.); sabrina.katzdobler@med.uni-muenchen.de (S.K.); guenter.hoeglinger@med.uni-muenchen.de (G.U.H.); 4Schoen Clinic München Schwabing, 80804 Munich, Germany; 5German Center for Neurodegenerative Diseases (DZNE) e.V., 81377 Munich, Germany

**Keywords:** pull test, test of retropulsion, postural instability, Parkinson’s disease, pose tracking, RGB-Depth sensor

## Abstract

Postural instability is a common complication in advanced Parkinson’s disease (PD) associated with recurrent falls and fall-related injuries. The test of retropulsion, consisting of a rapid balance perturbation by a pull in the backward direction, is regarded as the gold standard for evaluating postural instability in PD and is a key component of the neurological examination and clinical rating in PD (e.g., MDS-UPDRS). However, significant variability in test execution and interpretation contributes to a low intra- and inter-rater test reliability. Here, we explore the potential for objective, vision-based assessment of the pull test (vPull) using 3D pose tracking applied to single-sensor RGB-Depth recordings of clinical assessments. The initial results in a cohort of healthy individuals (*n* = 15) demonstrate overall excellent agreement of vPull-derived metrics with the gold standard marker-based motion capture. Subsequently, in a cohort of PD patients and controls (*n* = 15 each), we assessed the inter-rater reliability of vPull and analyzed PD-related impairments in postural response (including pull-to-step latency, number of steps, retropulsion angle). These quantitative metrics effectively distinguish healthy performance from and within varying degrees of postural impairment in PD. vPull shows promise for straightforward clinical implementation with the potential to enhance the sensitivity and specificity of postural instability assessment and fall risk prediction in PD.

## 1. Introduction

Postural instability is a common and disabling feature in advanced stages of Parkinson’s disease (PD) [1,2]. This phenomenon significantly contributes to recurrent falls [3,4,5], thereby exacerbating mobility issues, morbidity, and a decline in quality of life [6,7]. Additionally, postural instability marks a critical milestone in the progression of PD, indicating the onset of severe disability [2,8]. As such, it serves as a key element in assessing disease severity, forming an integral part of common rating scales such as the Hoehn and Yahr Disability Scale (HY) [9] and the Unified Parkinson’s Disease Rating Scale (MDS-UPDRS) [10].

The clinical gold standard for evaluating postural instability, particularly in PD, is the pull test, also known as the test of retropulsion. This test, incorporated into the UPDRS (UPDRS_PT_) [10], assesses balance-corrective responses by administering a sudden backward shoulder pull and observing the ensuing corrective steps or lack thereof. Despite its widespread use, the pull test suffers from considerable variability in execution and interpretation among examiners, leading to low inter- and intra-rater reliability [11,12,13]. Additionally, its coarse-grained rating scheme may further contribute to its limitations in accurately predicting important clinical outcomes, such as fall risk [13,14,15,16].

Efforts to mitigate the limitations associated with the pull test have led to the exploration of various technical strategies to standardize its execution and enhance its interpretive objectivity. Prior studies have examined different tools for this purpose. On one hand, attempts to standardize test execution have involved the use of external apparatus or force gauges to administer pulls at predetermined forces [17,18,19]. However, these approaches face challenges in clinical implementation and can only indirectly inform test execution, as the balance perturbation depends on the resulting acceleration of the patient’s center of mass, rather than the applied force itself. On the other hand, efforts to quantify test outcomes, such as stepping and truncal responses, have utilized elaborate laboratory-based equipment, such as force plates or marker-based multi-camera motion capture systems, which pose limitations in clinical implementation [18]. Alternatively, body-worn inertial sensors have been frequently employed [19,20,21,22], which can accurately monitor the temporal dynamics of the test (e.g., pull onset or step latency), but are prone to errors in quantifying absolute spatial parameters, such as step lengths or retropulsion angle.

In this study, we propose an alternative vision-based approach (vPull) to objectify the execution and assessment of the pull test in PD using markerless 3D pose tracking based on a single RGB-Depth sensor. This technique, previously validated for other types of clinical motor assessment [23,24,25], offers the promise of accurately monitoring test execution and providing detailed insights into balance-correcting responses that surpass the clinical rating scheme. Moreover, its low preparation and equipment costs make it suitable for straightforward clinical implementation. In the following, we will first evaluate the validity of vPull by assessing its agreement with a gold standard marker-based motion capture system in healthy subjects. Subsequently, in a cohort comprising PD patients and age-matched controls, we will assess the reliability of test outcomes and analyze the discriminatory power of quantified test responses in distinguishing between healthy individuals and patients with varying degrees of postural impairment.

## 2. Materials and Methods

### 2.1. Participants

Fifteen healthy subjects (age: 32.2 ± 7.0 years; height: 1.73 ± 0.12 m; weight: 71.9 ± 23.5 kg; BMI: 23.5 ± 4.6; eight females, seven males) participated in the first part of the study, which involved the validation of the vision-based measurement approach against the gold standard. In the second study part, the reliability and characterization of postural instability of varying degrees were examined in a cohort of 15 patients with PD (age: 70.1 ± 8.8 years; height: 1.69 ± 0.11 m; weight: 72.7 ± 12.7 kg; BMI: 25.3 ± 3.5; 8 females, 7 males) and 15 age-matched healthy controls (age: 69.9 ± 8.3 years; height: 1.73 ± 0.09 m; weight: 73.1 ± 11.1 kg; BMI: 24.2 ± 2.1; 7 females, 8 males). Prior to inclusion, healthy participants were screened for any neurological or musculoskeletal conditions that could affect postural control. Each participating patient underwent a complete physical, neurological, and neuro-otological examination by an expert neurologist (A.Z.). None of the patients showed any signs of atypical parkinsonism. Clinical scoring of disease stage and symptom severity revealed mild to moderate disease severity (HY of 2.0 ± 0.7 and MDS-UPDRS of 24.9 ± 14.4). L-DOPA was the basic medication in all patients (mean daily dose: 477 ± 239 mg). Regular medication was continued during study participation. Participants gave written informed consent prior to study inclusion.

### 2.2. Experimental Procedures

#### 2.2.1. Pull Test Assessment

The pull test was performed by two trained clinical examiners (N.E. and K.N.), which followed the instructions on the MDS-UPDRS form for conducting the pull test [10]. Participants stood erect with their eyes open and feet parallel to each other and comfortably apart. The postural perturbation consisted of a sudden and vigorous backward pull applied to the participant’s shoulders. After an instructional trial, each of the two examiners conducted two pull tests per participant.

#### 2.2.2. RGB-Depth Sensor Recording and 3D Pose Estimation

The RGB-Depth sensor (Azure Kinect, Microsoft, Redmond, WA, USA) was placed on a tripod, positioned at a height of one meter, two meters in front of the participant being examined (Figure 1A). Raw data (RGB resolution 3840 × 2160, Depth resolution 640 × 576, Depth FOV 75° × 65°) of the sensor was captured at a fixed sampling rate of 30 Hz by using the Kinect Azure sensor SDK (Version 1.4.1). Each recording had a fixed duration of 10 s, with both the start and end signaled by a sound signal.

The markerless 3D pose estimation was conducted in a two-step process. Initially, a 2D pose estimation was performed on the RGB stream of the sensor using a top–down approach. First, the participant was identified using a bounding box detector (YOLOv8 [26]), and then the entire pose (COCO 17 keypoints; Figure 1A) within the bounding box was predicted (RTMPose [27]). The 2D keypoints were then finally projected into 3D using the depth stream of the sensor.

#### 2.2.3. Marker-Based Motion Capture Recording

The validation of the vision-based measurement approach was performed with a marker-based motion capture system (Qualisys Medical AB, Gothenburg, Sweden) that consisted of nine wall-mounted cameras (Oqus series) that covered full-body motion at a sampling rate of 100 Hz within an area of 5 × 5 m. The recordings with the motion capture system were temporally synchronized with the RGB-Depth sensor. For the validation recordings, light-weight passive IR reflective markers with 19 mm diameter were placed on 36 anatomical landmarks to capture the motion of all major body joints as well as the head and trunk [23].

### 2.3. Data Analysis

For the analysis of the pulling movement and the subsequent stepping and truncal response, bilateral body landmarks of the shoulders, hips, and ankles were employed (Figure 1). Prior to further analysis, the trajectories of these 3D keypoints were smoothed using a 4th-order Butterworth low-pass filter with a cutoff frequency of 7 Hz [28].

The onset of the pull movement was estimated by the time when the acceleration of the shoulder keypoints exceeded a predefined threshold (3 SD above 1 s baseline at the beginning of the recording) [19]. Onset time was subsequently determined more precisely as the last local minimum preceding the resulting shoulder acceleration curve. The pull magnitude was determined as the peak of the resulting shoulder acceleration curve [21].

The stepping response was assessed using the 3D velocity profiles of bilateral ankle keypoints. Step initiation and termination were determined based on the time points where the ankle velocity surpassed or dropped below a predefined threshold (i.e., 0.7 m/s [28]), respectively. From the temporal sequence of steps, the following parameters were extracted: the latency of the first step (relative to the onset of pulling), the length and velocity of the first step, and the total number of steps. In addition to these established markers for characterizing the stepping response, we conducted an analysis of the truncal response to the postural perturbation. This involved evaluating the bending dynamics of the trunk in the backward direction following the onset of pulling. Specifically, we identified the maximum retropulsion angle relative to the baseline posture before pulling, utilizing bilateral shoulder and hip keypoints. Where applicable, we defined the time of balance recovery as the earliest moment when both stepping was terminated and the backward bending angle was reduced by at least 75% from its maximum after pulling.

In addition to the automated vision-based analysis of the pull test response, each participant’s response was clinically rated by an expert neurologist (A.Z.) through inspection of related video recordings, following the UPDRS_PT_ rating scheme. All data analysis procedures were performed in Python version 3.9.

### 2.4. Statistical Analysis

Descriptive statistics are presented as mean ± standard deviation (SD). Initially, we evaluated the agreement between the RGB-Depth sensor and the motion capture gold standard in estimating the described metrics that characterize the execution of the pull test and participant responses. For this assessment, only one pull test outcome per participant was considered, specifically the second pull test performed by examiner K.N. We employed multiple statistical techniques to assess agreement, including Pearson’s correlation coefficient, Bland–Altman analysis, and the intraclass correlation coefficient for absolute agreement (ICC (3,1); two-way mixed model). From the Bland–Altman analysis, we derived the bias (the average difference between measurement methods) and the reproducibility coefficient (RPC; 1.96 × SD of the difference between measurement methods) [29]. A non-zero bias indicates a systematic deviation between the methods, while RPC quantifies the overall variability between them. ICC (A,1) was interpreted according to the following established categories [30]: poor agreement (<0.5), moderate agreement (0.5–0.75), good agreement (0.75–0.9), and excellent agreement (>0.9).

Subsequently, we assessed the inter-rater reliability of RGB-Depth sensor-derived parameters in both PD patients and age-matched controls, considering two pull test outcomes per participant (specifically, the second pull test of the two examiners). Reliability was evaluated using (ICC (1,1); one-way random model) and the standard error of measurement (SEM), providing insight into the variability across repeated assessments.

Finally, we examined the ability of pull test outcomes obtained via our vision-based measurement approach to differentiate between healthy postural responses and impaired balance reactions across patients with varying degrees of postural impairment. Differences were analyzed using *t*-tests for numeric parameters and chi-squared tests for categorical parameters based on the outcomes of the second pull test performed by examiner K.N. We further evaluated potential correlations between the pull magnitude and the stepping and truncal response using Spearman’s rank correlation coefficient for categorical parameters and Pearson’s correlation coefficient for numeric parameters. We finally utilized Uniform Manifold Approximation and Projection (UMAP) to project the derived parameters into 2D space (considering the second pull test outcomes of both examiners). UMAP is a dimensionality reduction technique that helps to visualize high-dimensional data in a lower-dimensional space while preserving the underlying structure and relationships [31]. This allowed us to visually evaluate how effectively the test outcomes cluster, distinguishing between healthy and impaired balance responses across varying degrees of postural impairment. All statistical analyses were conducted using Python version 3.9.

## 3. Results

We validated our markerless 3D pose tracking approach, vPull, by comparing its performance in capturing pull test execution (i.e., pull magnitude) and balance-corrective response patterns (i.e., stepping and truncal response) of healthy participants against a gold standard motion capture system. Table 1 presents the detailed agreement statistics. Our analysis revealed overall good-to-excellent agreement for all derived parameters. Notably, the number of balance-corrective steps demonstrated the highest agreement, with an ICC (3,1) of 1, exhibiting zero bias and RPC. Conversely, the latency of the stepping response showed comparatively poorer but still commendable agreement, with an ICC (3,1) of 0.86. These findings affirm that vPull accurately captures the spatiotemporal execution and response characteristics of the clinical pull test.

In a cohort comprising patients with PD and age-matched controls, we subsequently evaluated the reliability of vPull by assessing the variability of derived test metrics within participants across repeated pull test assessments conducted by different examiners. Table 2 presents the detailed reliability statistics. While the clinical rating of test responses according to the UPDRS_PT_ scheme indicated excellent inter-rater reliability, with an ICC (1,1) of 1, significant variability was observed in the quantitative test metrics characterizing pull test execution and response. Specifically, the magnitude of the applied pull demonstrated only moderate repeatability, with an ICC (1,1) of 0.56. Similarly, the inter-rater reliability of test metrics characterizing truncal and stepping responses ranged from poor (e.g., latency of the stepping response, ICC (1,1) = 0.35) to good (e.g., first step velocity, ICC (1,1) = 0.78) reliability. These findings suggest considerable variability in pull test execution and spatiotemporal response patterns, which, however, do not significantly affect overall clinical categorization.

In a final analysis, we assessed the potential of vPull-derived test metrics to discern between healthy individuals and PD patients with varying degrees of balance impairment in response to the pull test (Figure 2A–J). We observed that examiners tended to apply a slightly softer pull to PD patients compared to healthy controls (*p* = 0.040). Variations in pull magnitude moderately influenced the stepping pattern, specifically the length (R = 0.45; *p* = 0.013) and velocity (R = 0.44; *p* = 0.017) of the first step but did not impact the overall clinical categorization according to the UPDRS_PT_ scheme. Clinical evaluation using this scheme indicated that all healthy participants exhibited normal responses, while approximately half of the patients demonstrated slight impairment (corresponding to UPDRS_PT_ grade 1; 53%), with the remaining showing moderate impairment (corresponding to UPDRS_PT_ grade 3; 47%). This distinction was reflected by quantitative test metrics. While all patients exhibited a corrective stepping response, only approximately half (53%) were able to regain balance without assistance. Detailed analysis of the stepping response revealed that patients required a higher number of corrective steps (*p* < 0.001), executed them more rapidly (*p* = 0.023), and with reduced velocity (*p* = 0.004). Detailed examination of truncal responses showed that patients exhibited a greater retropulsion angle (*p* = 0.014), and those able to recover took longer to do so (*p* = 0.004). Finally, low-dimensional embedding of all test metrics from automated video-based analysis via UMAP demonstrated distinct clustering of postural response patterns according to clinical ratings (Figure 2K). This suggests that vPull offers automated and effective differentiation between slightly impaired and healthy responses as well as between moderately impaired and slightly impaired responses.

## 4. Discussion

The central objective of this study was to introduce and evaluate a novel approach, vPull, aimed at standardizing and objectifying the clinical pull test for evaluating postural instability in PD and related movement disorders. vPull relies on a single RGB-Depth sensor, combined with state-of-the-art pose tracking technology, to monitor the 3D kinematics of patients during the execution of the pull test. Through comprehensive evaluation, we demonstrated an overall excellent accuracy and concurrent validity of our approach compared to the gold standard (i.e., marker-based motion capture system), encompassing metrics assessing both the magnitude of the pull and various spatiotemporal aspects of stepping and postural responses. Furthermore, within a cohort comprising patients and controls, we investigated the inter-rater reliability of vPull-derived metrics and illustrated typical disease-related impairments observed in postural responses of patients with PD.

Our approach enables us to confirm and extend previous findings related to pull test outcomes in clinical PD cohorts. By monitoring whole-body 3D kinematics during test execution, we quantify patients’ balance-corrective responses using established metrics, that primarily focus on stepping response characteristics [18,21], alongside newly introduced metrics characterizing truncal response dynamics. Consistent with prior research, vPull-derived metrics reveal a slowed and shortened stepping response in patients requiring a greater number of steps to regain balance (corresponding to UPDRS_PT_ grade 1) [18,21]. Moreover, patients frequently failed to sufficiently recover balance after initiating corrective steps, often requiring stabilization by the examiner (corresponding to UPDRS_PT_ grade 3). Cluster analysis on the complete set of vPull-derived metrics indicates a good separability between different UPDRS_PT_ grades, reflecting healthy responses versus increasingly impaired postural responses. Follow-up studies will have to explore whether the vision-based digital outcomes of the pull test possess the potential to more accurately predict clinically relevant endpoints, such as the risk of falling in PD [13,14,15]. Additionally, our continuous monitoring of whole-body 3D kinematics offers an opportunity to apply new deep learning approaches which may identify yet unknown aspects of impaired postural responses in patients, thereby making the clinical assessment of postural instability in PD potentially even more precise [32].

Consistent with findings from previous studies, we further observed significant variability in test execution between different examiners and within single patients across repeated assessments, resulting in an overall moderate to low inter-rater reliability [11,12,13]. Despite these variations, it is crucial to note that while changes in pull strengths influenced certain aspects of stepping response, particularly the size and speed of stepping [19,33], these variations remained within the category boundaries of the UPDRS_PT_ scheme and thus did not influence the overall clinical rating. Given that our approach provides direct feedback about the resultant destabilization magnitude, vPull holds promise for training clinical investigators in achieving a more uniform test execution in future. Beyond assessing pull magnitude, monitoring of whole-body 3D kinematics further offers the potential to characterize the dynamics of postural stability during the test more directly by continuously monitoring the relationship between the patient’s center-of-mass and base of support [34]. This in turn would enable immediate feedback to the examiner on the adequacy of test execution, i.e., whether the pull triggered postural instability or not, and could help establish a yet lacking standardization of the test execution, e.g., by defining a corridor between a just strong enough and an overly strong postural destabilization. Finally, monitoring the dynamics of postural stability throughout test execution would specify the precise temporal relationship between the onset of postural instability and the subsequent postural response—an aspect that has been suggested to be impaired in PD [18].

vPull offers inherent adaptability, enabling the monitoring of the pull test from different perspectives (also concurrently). From a practical standpoint, employing a single optical sensor for tracking is certainly preferable for clinical use. Both frontal and lateral perspectives offer unique advantages and disadvantages that warrant consideration. The lateral perspective, for instance, affords simultaneous monitoring of both examiner and patient behavior, facilitating the precise identification of potential stabilizing interventions by the investigator. However, it may encounter challenges in accurately capturing the stepping response due to potential occlusion of the stepping foot. Conversely, the frontal perspective allows for precise tracking of patients’ stepping responses but may only indirectly infer interventions by the examiner, relying on the observable truncal response of the patient. Given our primary interest on precisely characterizing patients’ balance-correcting response patterns, we here opted for the frontal perspective.

## 5. Conclusions

In conclusion, the vPull approach presents a potential avenue for standardizing the clinical pull test in assessing postural instability in PD and related movement disorders. vPull may contribute to establishing the pull test as an accurate and reliable digital biomarker that may inform disease staging, intervention monitoring, or fall risk assessment. Moreover, the utilization of a single low-cost RGB-Depth sensor for objective monitoring of test execution and response significantly reduces barriers to implementation, paving the way for widespread adoption in different clinical and research settings.

## Figures and Tables

**Figure 1 sensors-24-03673-f001:**
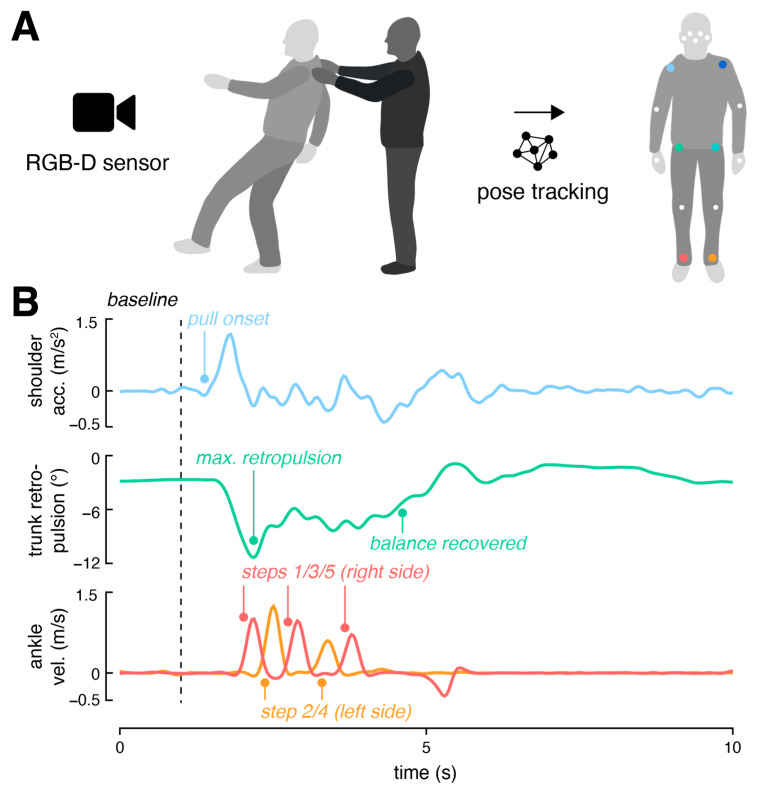
Experimental setup and analysis approach. (**A**) The pull test execution is recorded using a single RGB-Depth sensor positioned 2 m in front of the assessed patient. A 17 keypoint pose model is then estimated from the RGB frames and projected into 3D space based on the sensor’s depth frames. (**B**) Pull onset and magnitude are determined from 3D shoulder acceleration, utilizing thresholding relative to baseline shoulder motion. The amplitude of retropulsion and latency of balance recovery are assessed through 3D trunk bending dynamics in the backward direction. Steps are identified by thresholding 3D bilateral ankle velocities. *Abbreviations: acc: acceleration; vel: velocity*.

**Figure 2 sensors-24-03673-f002:**
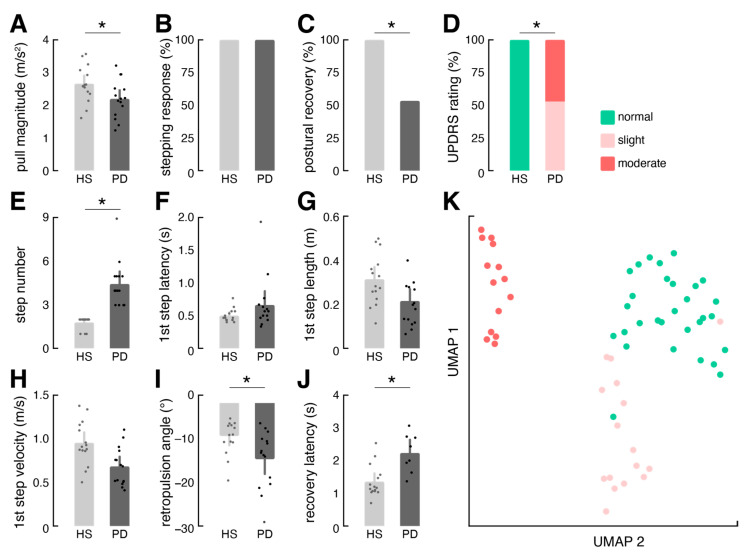
Group comparison of vPull test metrics between patients and controls (* indicate a significant difference between groups). (**A**–**D**) General test characteristics, including pull magnitude, percentage of individuals displaying a stepping response, percentage of individuals showing successful balance recovery, and corresponding rating of pull test performance according to the MDS-UPDRS scheme (grade 1—slight, grade 3—moderate). (**E**–**H**) Detailed metrics characterizing the stepping response. (**I**,**J**) Detailed metrics characterizing the truncal response and balance recovery. (**K**) Low-dimensional embedding of the above quantitative features (**E**–**J**) labeled by pull test performance rating using UMAP. *Abbreviations: HS: healthy subjects; PD: patients with Parkinson’s disease; MDS-UPDRS: The Movement Disorder Society Unified Parkinson’s Disease Rating Scale; UMAP: Uniform Manifold Approximation and Projection for Dimension Reduction*.

**Table 1 sensors-24-03673-t001:** Agreement statistics of derived pull test metrics of our markerless vision-based approach (vPull) compared to the gold standard (Qualisys).

Parameter	vPull	Qualisys	R	Bias	RPC	ICC (3,1)
Pull magnitude (m/s^2^)	2.98 ± 0.69	3.15 ± 0.64	**0.89**	−0.17	0.32	**0.93**
Step number	1.87 ± 0.71	1.87 ± 0.71	**1.00**	0	0	**1.00**
1st step latency (s)	0.57 ± 0.15	0.54 ± 0.09	**0.87**	0.02	0.09	**0.86**
1st step length (m)	0.32 ± 0.09	0.31 ± 0.09	**0.95**	0.01	0.03	**0.97**
1st step velocity (m/s)	1.12 ± 0.19	1.12 ± 0.16	**0.88**	0.01	0.09	**0.93**
Retropulsion angle (deg)	−10.52 ± 4.56	−11.39 ± 4.50	**0.92**	0.87	1.78	**0.95**
Recover latency (s)	1.67 ± 0.76	2.27 ± 2.56	**0.94**	0.12	0.29	**0.95**

Significant outcomes are marked in bold. Abbreviations: R: Pearson’s correlation coefficient; RPC: reproducibility coefficient; ICC: intraclass correlation coefficient.

**Table 2 sensors-24-03673-t002:** Inter-rater reliability statistics of vPull-derived pull test metrics.

Parameter	First Test	Second Test	SEM	ICC (1,1)
Pull magnitude (m/s^2^)	2.42 ± 0.59	2.39 ± 0.78	0.12	**0.56**
Step number	3.13 ± 1.73	2.97 ± 2.11	0.27	**0.73**
1st step latency (s)	0.58 ± 0.29	0.69 ± 0.59	0.10	**0.35**
1st step length (m)	0.27 ± 0.12	0.25 ± 0.14	0.02	**0.70**
1st step velocity (m/s)	0.82 ± 0.26	0.79 ± 0.30	0.03	**0.78**
Retropulsion angle (deg)	−10.66 ± 6.18	−10.45 ± 5.92	1.00	**0.61**
Recover latency (s)	1.66 ± 0.65	1.53 ± 0.61	0.12	**0.59**
MDS-UPDRS rating	0.96 ± 1.19	0.96 ± 1.19	0	**1.00**

Significant outcomes are marked in bold. Abbreviations: SEM: standard error of measurement; ICC: intraclass correlation coefficient; MDS-UPDRS: The Movement Disorder Society Unified Parkinson’s Disease Rating Scale.

## Data Availability

A sample dataset and the scripts for the pull test analysis used in this study are publicly available at https://github.com/DSGZ-MotionLab/vPull, accessed on 23 May 2024. The complete data from this study can be obtained upon reasonable request from M.W. Participants did not consent to the publication of their sensor data in open repositories, in accordance with European data protection laws.

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
