# Peer review of "Precision Balance Assessment in Parkinson’s Disease: Utilizing Vision-Based 3D Pose Tracking for Pull Test Analysis"

_sensors, 2024, doi:10.3390/s24113673_

Round 1
Reviewer 1 Report
Comments and Suggestions for Authors
This is a very intriguing study. The ability to quantify Parkinson's disease tests in a simplified methods is desired worldwide. This study has been appropriately designed, considering the differences in patient responses across tests, which is also characteristic of Parkinson's disease. We look forward to the development of a quantitative system that takes these factors into account.
Author Response
We would like to thank the reviewer very much for the positive feedback on our manuscript.
Reviewer 2 Report
Comments and Suggestions for Authors
Thank you for the opportunity to check out this interesting manuscript.
This approach enables to confirm and extend previous findings related to pull test outcomes in clinical PD cohorts. By monitoring whole-body 3D kinematics during test execution, quantify patients' balance corrective responses using established metrics,
that primarily focus on stepping response characteristics, alongside newly introduced metrics characterizing truncal response dynamics. Consistent with prior research, vPull-derived metrics reveal a slowed and shortened stepping response in patients requiring a greater number of steps to regain balance. From a practical standpoint, employing a single optical sensor for tracking is certainly preferable for clinical use. Both frontal and lateral perspectives offer unique advantages and disadvantages that warrant consideration.
To summarize, the utilization of a single low-cost RGB-Depth sensor for objective monitoring of test execution and response significantly reduces barriers to implementation, paving the way for widespread adoption in different clinical and research settings.
Author Response

(The authors gave the same response as above.)

Reviewer 3 Report
Comments and Suggestions for Authors
In this manuscript, the authors present a new approach to assess postural instability in Parkinson's disease. Overall, I think the article is very interesting and valuable. I present my comments and suggestions for changes in relation to the following parts of the article.
1. (Line 81, 85, 86) Are the 15 subjects mentioned in this paper 7 male and 8 female? Please also mention the number of male as well as female subjects. And, please also provide information about the subject's height, weight, and BMI.
2. The limitations of the study have not been mentioned. Please, add a limitation section.
Author Response
In this manuscript, the authors present a new approach to assess postural instability in Parkinson's disease. Overall, I think the article is very interesting and valuable. I present my comments and suggestions for changes in relation to the following parts of the article.
We would like to thank the reviewer very much for the positive feedback on our manuscript.
- (Line 81, 85, 86) Are the 15 subjects mentioned in this paper 7 male and 8 female? Please also mention the number of male as well as female subjects. And, please also provide information about the subject's height, weight, and BMI.
We have added the requested information about the study participants.
- The limitations of the study have not been mentioned. Please, add a limitation section.
We have addressed various limitations of the study in the discussion section and how these can be addressed in future studies. This includes investigating the predictive value of vPull results on clinically relevant endpoints such as fall risk, the currently missing implementation of actual stability criteria in the vPull analysis, and the use of deep learning methods to further improve the vPull analyses. We believe that repeating these points in a separate limitations section does not provide additional value to the reader.
Reviewer 4 Report
Comments and Suggestions for Authors
Pros.
The study presents the potential vision-based assessment of the pull test (vPull) using 3D pose tracking applied to single-sensor RGB- 24 Depth recordings of clinical assessment for healthy individuals and Parkinson’s disease (PD) patients appealing to the test of retropulsion, consisting of a rapid balance perturbation by a pull in backward direction that is regarded as the gold standard for evaluating postural instability in PD and a key component of the neurological examination and clinical rating.
The paper presents a methodology that refers to RGB-Depth sensor (Azure Kinect, Microsoft) and experiments supports the main idea.
Cons. The paper was submitted to a Special issue of "Wearable Sensors for Monitoring Athletic and Clinical Cohorts" but the Kinect could not be considered a wearable device. Is it?
The database is only available from the corresponding author upon reasonable request, so it can lead to low interest in other researchers.
Moreover, the only thing that is specified about the algorithmic development (beyond the statistical description) is that Python 3.9 was used, but what is the contribution when specifying the version of Python?, if there is no repository of the source codes as is usually done with other databases and repositories so that the community of researchers can replicate and verify or even improve the results, then this will surely lead to low interest from other readers.
Author Response
The study presents the potential vision-based assessment of the pull test (vPull) using 3D pose tracking applied to single-sensor RGB- 24 Depth recordings of clinical assessment for healthy individuals and Parkinson’s disease (PD) patients appealing to the test of retropulsion, consisting of a rapid balance perturbation by a pull in backward direction that is regarded as the gold standard for evaluating postural instability in PD and a key component of the neurological examination and clinical rating.
The paper presents a methodology that refers to RGB-Depth sensor (Azure Kinect, Microsoft) and experiments supports the main idea.
Cons. The paper was submitted to a Special issue of "Wearable Sensors for Monitoring Athletic and Clinical Cohorts" but the Kinect could not be considered a wearable device. Is it?
The editoral office suggested the submission of the manuscript to the named Special issue. The assignment could be reconsidered, if requested.
The database is only available from the corresponding author upon reasonable request, so it can lead to low interest in other researchers.
Our study participants, particularly the patients, did not consent to the publication of their sensor data in a public repository, in accordance with European data protection regulations. Therefore, we cannot publish the complete clinical dataset. However, we will make a sample dataset and all analysis scripts for the vPull evaluation available at https://github.com/DSGZ-MotionLab/vPull and included this information in the revised version of the manuscript.
Moreover, the only thing that is specified about the algorithmic development (beyond the statistical description) is that Python 3.9 was used, but what is the contribution when specifying the version of Python?, if there is no repository of the source codes as is usually done with other databases and repositories so that the community of researchers can replicate and verify or even improve the results, then this will surely lead to low interest from other readers.
See response to the above comment.
Round 2
Reviewer 4 Report
Comments and Suggestions for Authors
The authors have appropriately addressed the review comments.
The authors point out that they do not have the consent of the participants to share the data openly, and they provide a sample dataset and scripts, which is very accurate. Only comment, additionally, that they could apply some anonymization technique to share data, without revealing the identity of the participants, and verify that everything is still in accordance with current laws.